# The effect of poor vision on economic farm performance: Evidence from rural Cambodia

**Frederik Sagemüller** [ID] *, **Selina Bruns, Oliver Mußhoff**

Department of Agricultural Economics and Rural Development, University of Göttingen, Göttingen, Germany

* frederik.sagemueller@agr.uni-goettingen.de

## Abstract

Roughly one-fifth of the global population is affected by poor visual acuity. Despite the fact that inhabitants of rural areas in low-income countries are most distressed by this, no prior research has studied the impact of poor visual acuity on the economic performance of farms. We conduct a standardized eye test with 288 farm managers in rural Cambodia and find that around 30 percent of our sample suffers from poor visual acuity in terms of near-sightedness (myopia). Our analyses indicate a statistically significant and economically meaningful association of poor visual acuity with economic farm performance. Our results show that gross margins for cropping activities per year could be, on average, around 630 USD higher if farm managers were able to correct for poor vision. Our results suggest that poor visual acuity impairs farm managers from tapping the full potential of their business, which in turn decreases their chance to break the vicious cycle of poverty.

**Data Availability Statement:** All data and code files are available from the Mendeley data database. DOI: doi:10.17632/53x6gknpnh.1.

**Funding:** This work was supported by the Bundesministerium für Wirtschaftliche

## 1. Introduction

An estimated 596 million people are affected by mild, moderate or severe distance vision impairments and blindness [1]. To put the problem into an economic perspective, the total worldwide financial cost of visual impairments was estimated at three trillion USD [2] and the economic burden of uncorrected distance refractive error alone was estimated to be USD$202 billion per annum [3].

A growing body of literature is investigating occurrence, cause and consequence of visual impairments. With regards to occurrence, most disease burden is carried by low income economies of South Asia, East Asia and Southeast Asia [4]. Due to population growth and demographic change, a substantial increase in the prevalence of visual impairments is expected in the future, e.g. the global incidence of population with myopia will increase to roughly 50 percent by 2050 [3]. In terms of cause, cataract and uncorrected refractive error contributed to 55 percent of blindness and 77 percent of vision impairment in adults aged 50 years and older [5]. Uncorrected refractive errors like myopia and presbyopia are known to be among the largest causes for moderate and severe vision impairments, as well as blindness [5]. Looking at the consequences of poor visual acuity, studies show that uncorrected refractive error leads to a decrease in quality of life [6, 7]. This result is confirmed through a randomized control trial carried out in Bangladesh, Kenya and the Philippines which shows that cataract surgery to

Zusammenarbeit und Entwicklung, BMZ, Federal Ministry of Development Cooperation [grant number 81180342]. The funders had no role in study design, data collection and analysis, decision to publish, or preparation of the manuscript.

**Competing interests:** The authors have declared that no competing interests exist.

relieve blindness improves the quality of life of participants [8]. Moreover, people who underwent the surgery were more likely to participate in productive activities.

The losses in public welfare and human capital due to visual impairments are preventable; unaided visual impairments almost exclusively occur in the context of poverty. The relationship between poverty and eye health is laid out in a literature review in which the authors suggest that visual impairments and poverty seem to be intertwined in a vicious cycle [9]. A recent study shows that improving eye health contributes directly and indirectly to achieving many Sustainable Development Goals, including reducing poverty and hunger, improving work productivity and educational equity. The researchers suggest that eye health needs to be reframed as an enabling, cross-cutting issue within the sustainable development framework [10]. A combination of a lack of access to affordable public eye care in low-income countries, and the large percentage of the population in these countries whose livelihoods depend on agriculture, results in a greater burden of visual impairments occurring among inhabitants of agrarian communities. Thus, there are strong connections between agriculture, poverty and visual impairments.

From an agricultural economics perspective, we can hypothesize that poor vision is a cause for productivity loss of farms. Undoubtedly, the literature suggests that better public health contributes to higher agricultural productivity in terms of total factor productivity and efficiency [11–15], yet there are no case studies which describe effects of poor visual acuity on the profitability of farming. This research gap is astounding taking into account that raising agricultural productivity and profitability are central themes in the fight to end poverty [16–18].

The linkages between agriculture and visual impairments have only been directly addressed in one previous study, which shows that providing glasses to correct for age-related farsightedness (presbyopia) improves work performance of tea pickers in India [19]. The effect of myopia on the profitability of family-owned farms remains unknown. This aspect is important since agricultural production in developing countries relies mostly on family-owned farms [20] and the effect of vision impairments on economic performance could be biased if studied only among wage earners. Also unknown is the effect of myopia on agricultural profitability, as compared to the effect of presbyopia on work performance. Age-related presbyopia occurs in persons older than 40 years and therefore it is difficult to generalize the findings from their study.

We contribute to the understanding of the relationship between health and agriculture by adding a case study with farm managers in rural Cambodia on poor visual acuity in terms of myopia and its association with farm profitability. We address this topic by answering two research questions: 1) What is the prevalence of myopia among rural smallholders in Cambodia? 2) Is myopia associated with a loss in agricultural profitability of family-owned farms?

To illustrate the relationship between agricultural profitability and poor visual acuity, and to give first estimates on potential effect sizes, we carry out a household survey combined with a standardized eye test with 288 Cambodian smallholders. Cambodia was chosen for our empirical application for three reasons: Firstly, the anecdote that the leaders of the Communist Party of Kampuchea (CPK) 'decided to kill anyone who wore glasses', is commonly shared when relaying some of the horrors associated with life in Cambodia during the revolutionary period [21]. It is plausible that cultural stigmas and stereotypes are playing parts in the low uptake of glasses in rural Cambodia. Secondly, the rural areas of Northeast Cambodia are coined by slow rural development. Most income is generated by small scale agriculture and most of the economically active population is employed in and depends on agriculture. Thirdly, the predicted increase in the number of people with avoidable vision impairment to 2050 is mainly occurring in South Asia and East Asia [5].

This paper is addressed to development practitioners and researchers that work in agriculture and global health. There is a growing recognition that opportunities exist for agriculture to contribute to better health, and for health to contribute to agricultural profitability. We argue that joint action in agriculture and health could unlock synergies that substantially reduce poverty. From a research perspective, our study gives first results on the association between poor visual acuity and agricultural profitability and we want to motivate research at the intersection of public health topics and agriculture. The implications of our results are critical due to the magnitude of people who suffer from poor visual acuity and have limited access to modern optometric services. The issue becomes more pressing when we consider that the number of people affected by uncorrected poor visual acuity will continue to rise in the future [3].

The following paper is structured as follows: Section 2 reviews the literature on the global losses of productivity due to visual impairments with a special attention to the agricultural sector. Section 3 lays out a very simple conceptual framework to describe how visual acuity affects agricultural profitability. Section 4 gives an overview on the data collection process, the raw data, the important variables we include in our model and the empirical strategy. Section 5 presents the results and discussion. In Section 6 we draw a conclusion.

## 2. Literature review

Vision is often considered to be the sense that is most valued [22]. Vision requires structural and physiological integrity of the eyes, brain, and their connections. Disruption of any part of this pathway causes vision impairment. The most common causes of vision impairment in adults are uncorrected refractive error, cataract, glaucoma, age-related macular degeneration, diabetic retinopathy, corneal scarring, and trachoma. Vision-driven activities of daily living can be captured using quality of life tools and vision function-related tasks. The most common measure of visual function is distance visual acuity, which tests the ability to discern letters or characters of high contrast at decreasing size using the central retina [23, 24].

In 2020, an estimated 510 million people worldwide, of whom most live in low and middle income countries, had uncorrected near vision impairment, and a further 596.2 million people have distance vision impairment [23]. Due to population growth and demographic change, a substantial increase in the prevalence of visual impairments is expected in the future, e.g. the global incidence of population with myopia will increase to roughly 50 percent by 2050 [3]. An important gap in this literature is the lack of data from low income regions, including Southeast Asia. Due to this lack of data, most prevalence based studies extrapolate estimates across regions. Anyhow, there are studies that report regional estimates for South East Asian countries, though most of them report data on the sub-national level, within specific age groups and on a variety of different indicators for visual impairments. For example, a study from rural Myanmar looks at the prevalence of refractive errors in a population cohort of 40 years and older. The study reports a prevalence of refractive errors of 42.7 percent [25]. A national survey from Thailand reports an incidence of refractive errors via self-assessment, which is reported to be 28 percent [26]. A study from an urban area in Lao PDR conducts comprehensive ophthalmic examinations and finds that the incidence of bilateral visual impairments of the population was 22.4 percent [27]. A survey from Cambodia reports a prevalence of low vision in adults 50 years and older to be 21.1 percent [28]. The prevalence of vision impairment in school children between the ages 12–15 in Vietnam is reported to be 19.4 percent [29]. Despite the methodological differences and the differences of the cohort populations in age and in rural and urban locations, these studies show that the prevalence of low vision in the region ranges around a fifth to a quarter of the population.

From a macroeconomic perspective, there is evidence that vision impairments have a large economic impact worldwide. The scientific literature heavily relies on studies that calculate welfare costs of vision impairments by using visual acuity prevalence reported in national and global datasets, together with data for relative reduction in employment and reduction in wages due to visual impairments. Anyhow, the underlying data, methods and measurements of visual impairments have undergone drastic changes in recent decades. In 1996, the first global estimate of the worldwide productivity cost of blindness was estimated at $168 billion using 1993 data on visual impairment prevalence rates, gross domestic products and world population data [30]. The weakness of this study is that it only accounts for blindness as a visual impairment, and that the researchers assumed zero productivity for the blind and 100 percent productivity for the non-blind. A further study used data from the year 2000 to identify the potential effect on the global economic productivity of interventions that were planned as part of the "VISION 2020- right to sight" initiative [31]. The economic gain of the interventions was estimated at 102 billion USD. These results were rather conservative estimates, as admitted by the researchers, since it was assumed that only working individuals at working age (15–64 years) produce goods and services valued at GDP per capita. Another important weakness of these studies is that they only account for best corrected visual acuity. However, using best corrected visual acuity obscures that, especially in settings of low income economies, people may not own spectacles, and so live with vision impairment from uncorrected refractive error. This underestimation is possibly large, the total number of persons with visual impairment worldwide including uncorrected refractive error was estimated to be 61 percent higher than the commonly quoted estimates which exclude uncorrected refractive error [32]. Another estimate of the loss in productivity for 7 world regions was published in 2012. The study reports losses to be 168.3 billion USD, with projections for the year 2020 to be 177.5 billion USD [2]. Nevertheless, since losses to the economy were only accounted for in high income regions, these estimates are also underestimating the global costs due to losses in productivity. A recent study estimates the annual potential productivity losses associated with reduced employment due to blindness, moderate and severe vision impairment at the regional and global level. In this study, it is estimated that globally, 160.7 million people with moderate or severe vision impairment or blindness were within the working age. The relative reduction in employment by people with vision loss was 30.2% which result in a global potential productivity loss due to vision impairments of 410.70 billion USD purchasing power parity [33]. However, the study captures only a limited amount of productivity loss components. Components not included in the analysis, because reliable data at country and regional level remain scarce, were absenteeism and presenteeism (reduced productivity in the working place), premature mortality due to visual impairments, productivity losses of people older than 64 years, productivity losses of caregivers, and value of time lost from unpaid or informal labor activities.

Thus, what remains unknown from these estimates, are the losses that occur in the informal agricultural sector due to visual impairments. Therefore, even though these estimates are the best guess of global productivity losses to date, they most likely present a conservative estimate. Regarding the underlying mechanisms that would lead to losses in agricultural productivity, the existing scientific literature provides consistent evidence for an association of visual impairments with reduced quality of life [34], reduced educational outcomes [35], reduced social status and reduced economic activity [36] and cognitive impairment, cognitive decline, and dementia [23]. Other health domains and their association with agricultural productivity received more attention in the literature compared to visual impairments. For example, it is documented that health care access has a positive impact of total factor productivity for aggregate U.S. agricultural production [15] and that the general health status of Filipino, Malian,

Nigerian and Norwegian farmers increases production efficiency [12, 14, 37] and labor productivity [13]. The linkages between agriculture and visual impairments have only been directly addressed in one previous study, which shows that providing glasses to correct for age-related farsightedness (presbyopia) improves work performance of tea pickers in India [19]. The effect of myopia on the profitability of family-owned farms remains unknown. This aspect is important since agricultural production in developing countries relies mostly on family-owned farms [20] and the effect of vision impairments on economic performance could be biased if studied only among wage earners. Also unknown is the effect of myopia on agricultural profitability, as inference cannot be drawn from the effect of presbyopia on work performance. Age-related presbyopia occurs in persons older than 40 years and therefore it is difficult to generalize the findings from their study.

## 3. Conceptual framework

Our conceptual framework is related to the work on general impacts of health on economic outcomes, which describe how healthier populations tend to have higher labor productivity, because their workers are physically more energetic and mentally more robust [38]. Healthier children learn and perform better at school which leads to greater productivity and higher incomes. Furthermore, good health promotes school attendance and enhances cognitive function. We take a qualitative report from rural dwellers in Nepal as a starting point to map out the potential effects of poor vision on agricultural profitability [6]. The report explores the impact of corrected and uncorrected refractive error on Nepalese people's quality of life. We sort the qualitative statements from their report and group them by categories that affect the outcome we aim to study. Fig 1 shows that we expect strictly negative impacts of poor visual acuity on agricultural profitability. To describe the pathways from visual acuity to agricultural profitability, we order the qualitative statements into the following categories: 1. General ocular limitations. This category includes blurred vision and vision problems in general, like sensitivity to bright or dim light and limitations regarding reading and writing as well as riding a motorcycle. 2. Agricultural activity limitation. This category is specific to limitations in field work, like problems in seeing small insects, harvesting or using hand tools. 3. Limitation in access to information. This category entails limitations like reading newspapers, using a computer, reading calendars and clocks, as well as using a phone. 4. Physical discomfort symptoms are grouped and entail examples like squinting, a loss of balance or a pain in the eyes. 5. Limited social interactions. This entails examples of how people avoid crowded spaces, meeting people, attending social functions or recognizing faces. 6. Psychological symptoms and limitations. This refers to feelings of worry and depression, as well as nervousness and fears. 7. Limitations in business administration. Examples include making bank transfers, signing documents and recharging credit on the mobile phone. A detailed list of all limitations that are mentioned in the original study is given in the S1 File. This paper does not examine the effects of poor vision on all individual categories, but tries to estimate their aggregate effect.

Regarding the condition of poor visual acuity in Fig 1, poor visual acuity can be assessed by testing the performance of different components of the visual system. There are visual function tests that assess factors such as visual acuity, contrast sensitivity, color, and depth and motion perception. These properties each represent an aspect of visual function and impact an individual's level of functional vision. Visual acuity is the main component tested to assess the performance of the visual system and it is arguably the most crucial component of the visual system when it comes to working in agriculture [24]. Therefore, our study focuses on visual acuity as a proxy for visual function.

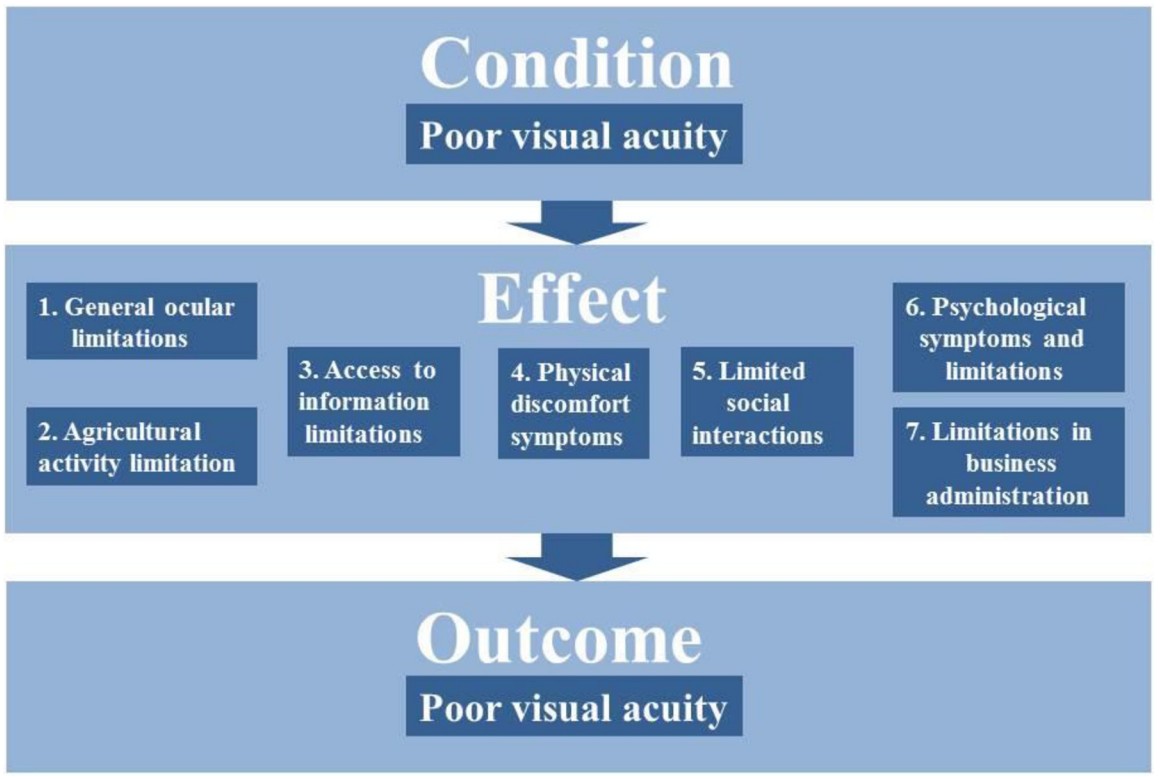

**Fig 1. Poor visual acuity and its negative effects on agricultural profitability.** The effects are derived from Kandel et al. (6), themes 1, 3, 4, 5, 6 and 7. Theme 2 was excluded because it deals exclusively with the negative effects of wearing glasses, contact lenses and corrective surgery. Source: Own depiction.

## 4. Methods

### 4.1. Ethics statement

When we conducted the fieldwork for this project in 2018, the University of Göttingen only had an internal review board system in place for clinical trials. Since our research does not qualify as a clinical trial, we were not eligible for an internal review. Anyhow, in close collaboration with our partners at CIAT (The International Center for Tropical Agriculture) and the Royal University of Agriculture in Phnom Penh, we designed this research project according the principles of ethic responsibility in research involving human subjects and the national legislation of Cambodia. Since 2020, the University of Göttingen has an ethics committee that reviews and approves research that involves human subjects. We submitted our research protocol to them and obtained a retrospective approval.

Before visiting the villages, we met the commune leader and the government extension officers of the respective sub-province. The extension officer then accompanied us to each village and introduced us to the village chief, whom we presented our research endeavor in order to get his/her consent for data collection in his/her village. Only after receiving verbal consent from a) the commune chief, b) the extension officer, and c) the village chief were we able to undertake the data collection in the respective village. Before the survey, we explained to each participant what the research is about, what their participation in the project entails and that participation is voluntary. After this was understood we gathered written consent from each participant to be included in the research project. As a compensation for their lost time, the

participants where paid equal to half a day of paid labor. We employed students from the Royal University of Agriculture who spoke English and Khmer to enable us to communicate with the farmers as well as other local stakeholders. We carried out intensive training sessions on survey methods which included the importance of explaining our research and obtaining informed written consent. One researcher was always present during data collection and she checked every questionnaire and if the protocol for obtaining consent was followed. The data was recorded by paper based surveys. The finished questionnaires were transcribed to an excel table by the researchers with codes for each observation so that re-identification without the paper based survey is not possible. The paper based surveys are archived at the University of Göttingen and only the authors of this article have access to them.

## 4.2. Data and descriptive statistics

We collect and explore cross-sectional primary data from household surveys and standardized eye examinations with 288 smallholder farmers from 16 villages throughout Ratanakiri province, Cambodia. The data was collected between August and October 2018 by a team of student enumerators from the Royal University of Agriculture in Phnom Penh, the University of Göttingen as well as staff from the provincial department of agriculture. We trained the enumerators and accompanied them during the data collecting process and conducted data quality checks. Ratanakiri province is remotely situated in Northeastern Cambodia. This multi-ethnic province is categorized as one of the poorest areas in Cambodia [39]. Of its 150,000 citizens, 88 percent live in rural areas and depend predominantly on income from agriculture. Rice is typically cultivated for household consumption whereas cassava, cashew, and rubber are the main cash crops [40–42]. The target villages were selected by the managers of a greater project on sustainable farming practices in the region. Since there are no comprehensive lists of farming households in the villages, we relied on the expert knowledge of the extension workers from the regional government offices and the respective village officials to select participants based on a nonprobability sample [43]. The household surveys recorded detailed data on crop production for the growing season of 2017–2018. Observations with missing values were dropped from the data set. The final data set contains 260 observations. Table 1 displays the variables that are used in the estimations and their precise measurements.

To assess visual acuity of participants, we carry out a standardized eye examination, the Landolt–C test. This test is particularly useful to test young, illiterate or non-English speaking populations for visual acuity [44]. The participants of the test go through six lines of a vision chart, with each line corresponding to a value of visual acuity. The Landolt rings get smaller in each line so it becomes increasingly difficult to identify the rings and the test is stopped when the participant cannot recognize the rings anymore and the last ring that was identified is recorded as the result. Following the standard test procedure, there are 22 rings and 7 visual acuity groups: VA = 0, VA = 0.3, VA = 0.4, VA = 0.63, VA = 0.7, VA = 1, VA = 1.25. The test is carried out first with the left eye, while blocking the right eye. Afterwards the test is repeated with the right eye while blocking the left eye. Also, the test is carried out without vision aids like glasses or contact lenses. In the following we use average VA score for both eyes. We also carry out calculations by taking the VA score for the weaker and stronger eye separately. The results do not differ from the results presented here. Fig 2 shows the raw results from the eye examination. Almost 5 percent of participants didn't identify a single ring and almost 20 percent of participants identified all 22 rings.

We use the visual acuity threshold of 0.7 in the decimal notation (visus) to classify participants into two groups: "poor vision" and "good vision". We use this threshold because it is a widely accepted indicator for the performance of the visual system, i.e. it is used to verify if a

**Table 1. Data description for selected variables.**

| Variable | Description |
|---|---|
| Gross margin | All produce valued at average product prices minus cost for seeds, fertilizer, insecticides, fungicides, herbicides, machine hours, land and costs for hired labor for all cropping activities (transplanting, weeding, application of agrochemicals, harvesting and irrigation). Relates to the growing season 2017–2018. All values are transformed to USD/year |
| Single factor productivity | Calculates revenues per farm and year, divided by the area under cultivation for the growing season 2017–2018. All values are transformed to USD/ha/year |
| Eyesight | Calculates the results from Landolt C-Test, classifying respondents into "poor vision" and "good vision". The threshold is an average visus on both eyes≥0.7 |
| Eyesight: Upper bound comparison | We shift the threshold of assignment to the "good vision" group to a visus≥0.75 |
| Eyesight: Lower bound comparison | We shift the threshold of assignment to the "good vision" group down to a visus≥0.45 |
| Age | Age in years |
| Area of cultivation | Total area of cultivation in hectares for all plots that belong to the farm |
| Education | Years in school |
| Household size | Number of people living in the household |

Source: Own data.

person's visual function is well enough to safely operate a vehicle [24]. In the context of our study, it is a suitable indicator because it applies a measure of visual acuity to a visual function. Thus, our indicator corresponds to visual functionality in everyday tasks, which connects to the idea of disadvantages in farm management activities for people who belong to the poor vision group. Another reason for selecting the two categories is that in practice, these categories resemble a real-world treatment. If we would give glasses to a person from our experiment, they could (theoretically) switch instantly from the poor vision group to the good vision group. Fig 3 displays the difference in test scores between the two groups. The poor vision group has an average VA score of 0.41 and the good vision group has an average VA score of 1.01.

From the household survey we collect production data on the growing season of 2017/2018. We record data on all arable crops of the household farm. In total, we recorded data from 543 individual plots on 260 farms. To aggregate the production of all commodities into a single measure, all produce is valued at average farm gate price in USD per year. Revenues here are not farm income, because not all produce is sold in the market; a large quantity of fruits, vegetables and rice is consumed by the household.

For each farm, we calculate the gross margin (USD/year) for the growing season 2017–2018:

$$Gross\ Margin(USD/year) = Revenues - (Labor\ costs + Input\ costs + Land\ rent)$$

Table 2 displays the gross margins. Input costs include expenditure for fertilizer, pesticides, fungicides and insecticides, seeds, planting materials and hired labor for all cropping activities and rent for land, where we apply average land prices per hectare. The overview statistics are displayed in Table 2 for the good vision group and the poor vision group.

The mean value of revenues for the good vision group is 2,265 USD per year and 2,058 USD per year for the poor vision group. In terms of input costs, the poor vision group has higher costs of hired labor, higher input costs and higher land rent costs on average. The

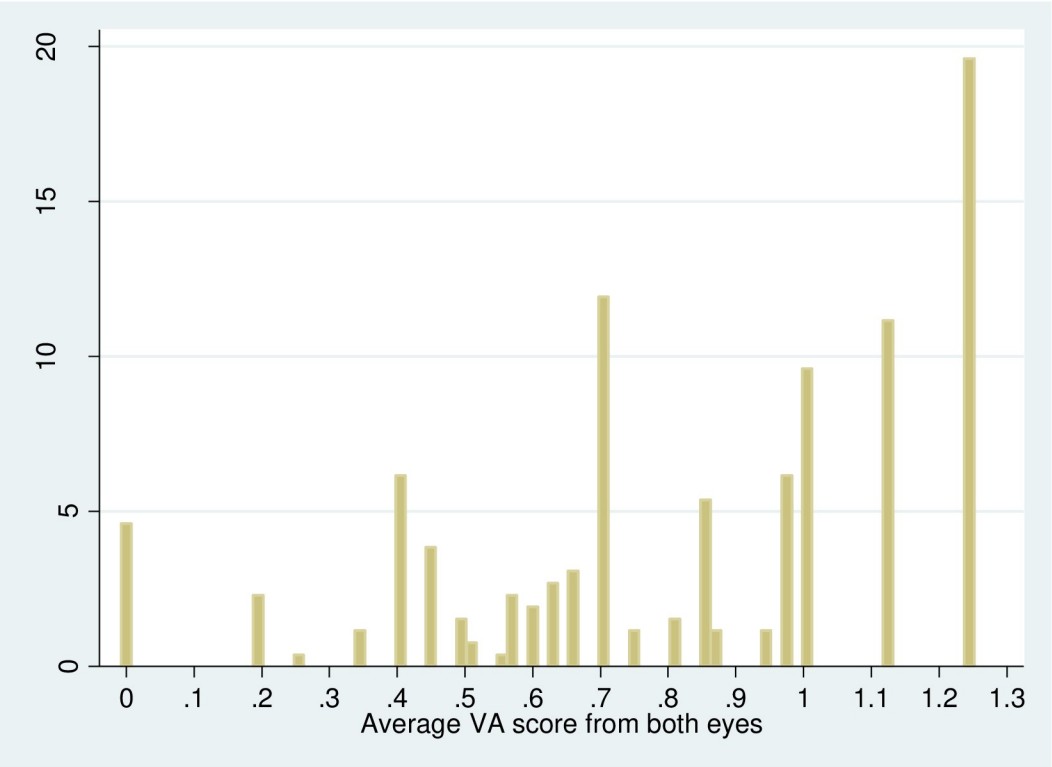

**Fig 2. Raw results from the standardized eye examination.** Number of observations = 260. Displayed are the average values from both eyes. Source: Own depiction.

resulting contribution margins are higher for the good vision group with 1,466 USD per year on average, compared to the poor vision group with 934 USD per year on average.

Table 3 shows a mean comparison of the independent variables for the two groups. The most important point here is the variation in age between the two groups. In the good vision group, participants are on average 33 years old, compared to the poor vision group with an average age of 49 years. This negative relationship of age with visual acuity is to be expected, because visual acuity is fully developed at about 12 months of age and decreases over time [45]. We carry out a t-test and see that this difference is statistically significant. Another variable with a statistically significant difference between the groups is area of cultivation, where the good vision group has on average 3.17 ha of arable land, compared to the poor vision group with 4.20 ha.

### 4.3. Empirical strategy

According to the conceptual framework in Fig 1, this paper explores the effect of poor vision of farm managers on the profitability of agriculture. An important problem of causal inference is how to estimate treatment effects in observational studies, where (like an experiment) a group of units is exposed to a well-defined treatment, but (unlike an experiment) no systematic methods of experimental design are used to maintain a control group [46]. To circumvent this problem, we apply Mahalanobis distance matching (MDM) and propensity score matching (PSM). Matching involves pairing treatment and comparison units that are similar in terms of their observable characteristics. These matching methods have become popular in impact

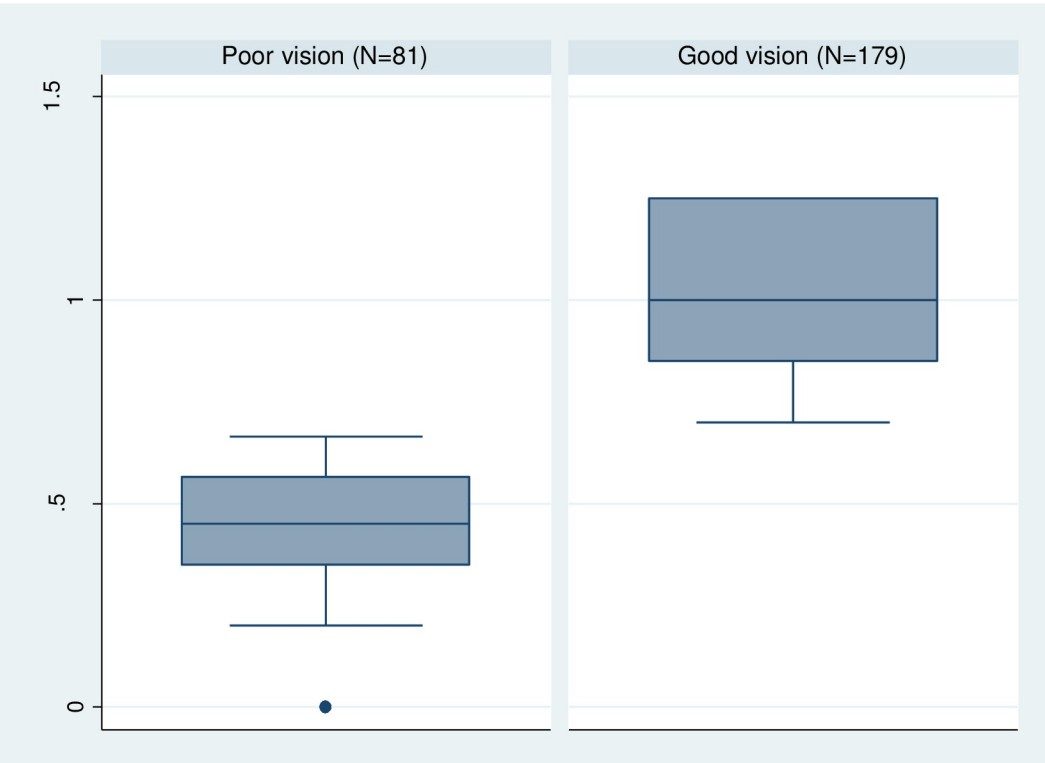

**Fig 3. Results from the standardized vision test by visual acuity group.** Number of observations = 260. Source: Own depiction.

evaluations and are used in a variety of fields, including to assess impacts related to agricultural production [47–51]. Admittedly, the ideal data to answer our research questions would come from a controlled experiment, where agricultural profitability is quantified before and after treating poor vision, for example by giving out glasses to the participants. In our case, an adequately powered randomized control trial is not feasible due to reasons such as ethic concerns, time and cost. Instead, we apply a less expensive strategy to explore observational data that is naturally occurring in the field. This way, we apply a cost-efficient analysis to generate first results on the topic to motivate further research. Two conditions about remote areas in

**Table 2. Calculations of contribution margins.**

|  | Good vision | | | | Poor vision | | | |
|---|---|---|---|---|---|---|---|---|
|  | Mean | S.E. | 95% CI | | Mean | S.E. | 95% CI | |
| Revenues[1] | 2,265 | 314 | 1,648 | 2,883 | 2,058 | 218 | 1,629 | 2,486 |
| Labor cost | 129 | 23 | 83 | 175 | 199 | 41 | 118 | 281 |
| Input cost | 164 | 23 | 120 | 209 | 255 | 45 | 166 | 344 |
| Land rent | 514 | 33 | 450 | 578 | 674 | 51 | 574 | 774 |
| Gross margin | 1,466 | 296 | 883 | 2,050 | 934 | 163 | 613 | 1,255 |

[1]All calculations are based on cost benefit analysis of 260 farms for the growing season 2017/2018.

Calculations are based on cropping activities on 543 single plots. Crops are cashew, cassava, fruits, maize, rice (upland), rubber, soybean and vegetables. All output is valued at average product price at farm gate in USD/year. Source: Calculated by the authors.

**Table 3. Mean comparison of key variables for the good vision and poor vision groups.**

| Variable | Mean | | T-test | |
|---|---|---|---|---|
| | **Good vision** | **Poor vision** | **T-value** | **p>|t|** |
| Age (years) | 33 | 49 | -10.63 | <0.01*** |
| Area of cultivation (ha) | 3.17 | 4.20 | -3.18 | <0.01*** |
| Education (years) | 3.05 | 2.56 | 1.44 | 0.15 |
| Gender (female = 1) | 0.56 | 0.61 | -1.00 | 0.32 |
| Household size (people) | 5.33 | 5.17 | 0.62 | 0.53 |

Significance levels:

* p<0.10,

** p<0.05,

*** p<0.01,

N = 260. Source: Calculated by the authors.

Southeast Asia enable this strategy: 1) a high and near-random incidence of "poor vision" among the target population and 2) the symptoms of poor vision are not treated by vision aids such as spectacles or contact lenses, which allows for a clear identification of treatment. Thus, we apply an estimation strategy that resembles a natural field experiment with regards to the near-random assignment to the group of poor vision and good vision. The strongest confounders in our data are observable, and we control for them in our estimations. This approach is restricted to the rural areas of most low-income countries, or anywhere the incidence of poor vision is high and health care infrastructure low [52].

We cannot measure the effect of belonging to the good vision group on agricultural profitability for each individual because we can only observe one outcome for each individual. Therefore, the focus of our analysis is on the average or population treatment effects, by using a potential outcome approach [53]. In our case it makes sense to investigate the average treatment effect on the treated (ATT), to explicitly evaluate the effects on the population for which the intervention is intended. This way we can estimate the realized gross gain that individuals get from having at least intact visual functions in terms of visual acuity. Put simply, our treatment effect resembles a possible real-world scenario, which would be a benchmark for the realized gross gain of an intervention to remedy poor visual acuity by prescribing glasses or contact lenses. The average treatment effect on the treated $\tau_{ATT}$ of our population is defined as follows:

$$\tau_{ATT} = E(\tau|D_i = 1) = E(Y_{1i}|D_i = 1) - E(Y_{0i}|D_i = 1), \tag{1}$$

where $D_i$ is a binary variable equal to one if participant $i$ passes the threshold of the vision test, zero otherwise, and $Y_{1i}$ and $Y_{0i}$ are the outcomes of the individuals with good vision and poor vision, respectively. The last term on the right-hand side of the equation is not observed, since it describes the hypothetical case of the outcome without treatment for the treatment group. To continue our estimation and find the true parameter for $\tau_{ATT}$ we replace the last term in formula (1) $E(Y_{0i}|D_i = 1)$ with $E(Y_{0i}|D_i = 0)$ so that $E(Y_{0i}|D_i = 1) - E(Y_{0i}|D_i = 0) = 0$.

To do this, we apply MDM and PSM We apply multiple matching methods, because it is recommended to use of several matching methods in combination with diagnostic checks to make a sample robust to the failures of individual methods [54]. We create the missing counterfactual from the pool of observations in the poor vision group by observable characteristics $x_i$, which is highly dimensional. To reduce the problem of multidimensionality in matching,

we match on a single index, the propensity score [55]. Matches are constructed on the basis of observed characteristics $x_i$ of the poor vision group and the probability to belong to that group $Pr(D_i = 1|x_i) = P(x_i)$ [55]. In the case of the MDM we calculate the inverse of the covariance matrix for all the covariates. Now, we can express the $\tau_{ATT}$ as:

$$\tau_{ATT}P(x_i) = E[Y_{1i}|D = 1,\ P(x_i)] -\ E[Y_{0i}|D_i = 0,\ P(x_i)]. \tag{2}$$

We use two different PSM estimators to obtain the results, the first being Kernel Based Matching (KBM), the second being Nearest Neighbor Matching (NNM). KBM averages over multiple individuals in the poor vision group for each individual in the good vision group, with weights defined by their distance [56], NNM is a one-to-one matching method where observations from the good vision group are assigned their closest match from the poor vision group. A major advantage of KBM is a lower variance, which is achieved because information from more observations is used. A drawback of the KBM method is that observations are possibly used that are bad matches [57].

To assess matching quality, a balancing test is required. The algorithm we apply splits a sample into equally spaced intervals of propensity scores and then tests whether average propensity scores between treated and control units are different [58]. Tests continue until the average propensity scores of the good vision group and propensity scores of the poor vision group do not differ in each interval. If the means of each characteristic between the good vision group and the poor vision group for the same propensity score do not differ, the balancing test is satisfied. We restricted the algorithm to test in the area of common support (the area belonging to the intersection of the propensity score of good vision and poor vision), as this condition enhances the quality of matches in ATT estimation.

In terms of sensitivity analysis, we apply a method that can reveal robust baseline results by comparing our results with a model that includes a binary variable that is a proxy for a potential unobserved confounder [59]. This potential confounder can be simulated in the data and used as an additional covariate in combination with the preferred matching estimator. The comparison of the estimates obtained with and without matching on the simulated confounder show to what extent the baseline results are robust to specific sources of failure of the conditional independence assumption.

## 5. Results and discussion

### 5.1. Main results

In Table 4, the results from PSM are displayed. As expected, the effect of the variable *Age* is statistically significant and negatively correlated with our binary treatment variable. We restrict the model to the region of common support and 255 of 260 observations are in the region of common support (176 observations from the good vision group and 79 from the poor vision group). The model has a high degree of sensitivity (92.18 percent) and specificity (60.49 percent). For both, negative and positive predicted values, the model correctly classifies 82.31percent of observations.

In Table 5 we compare the matched and unmatched samples. We use several parameters to assess matching quality. Apart from the mean values, we calculate the standardized differences of the original and matched samples. We can see that for the variable *Age*, matching reduces the standardized differences between the two samples to 18.00 percent, down from -133.80 percent in the original sample. The next highest standardized difference in the matched sample is reported for the variable *Education* with 13.70 percent. The test statistics show that the Rubin's B value, which is the absolute standardized difference of the means of the propensity score between the two groups, is 26.00 in the matched sample compared to 130.10 in the

**Table 4. Estimates from the PSM (treatment = good vision) with a probit model.**

| Variable | Coefficient | Standard error | Z-value | P>Z |
|---|---|---|---|---|
| Age (years) | -0.06 | 0.01 | -7.39 | <0.01*** |
| Area of cultivation (ha) | -0.38 | 0.03 | -1.18 | 0.23 |
| Education (years) | -0.04 | 0.03 | -1.16 | 0.24 |
| Household size (number of people) | 0.08 | 0.04 | 1.85 | 0.06 |
| Number of observations | 260 | | | |
| Sensitivity (%) | 92.18 | | | |
| Specificity (%) | 60.49 | | | |
| Positive predictive value (%) | 83.76 | | | |
| Negative predictive value (%) | 77.78 | | | |
| Correctly classified (%) | 82.31 | | | |
| LR chi2(7) | -18.40 | | | |
| Pseudo $R^2$ | 0.27 | | | |
| Observations on support (treatment) | 176 | | | |
| Observations on support (control) | 79 | | | |

Significance levels:

* p<0.10,

** p<0.05,

*** p<0.01.

Sensitivity is the ratio of predicted positives/ actual positives and specificity is the ratio of predicted negatives/ actual negatives. Source: Calculated by the authors.

unmatched sample. This is slightly higher than the value of 25.00, which is an indicator for good matching quality [60]. Additionally, Rubin's R value gives the ratio of the treated to control variances of the propensity scores. The value of our matched sample is 1.64, which is in the satisfactory range between 0.5 and 2.0 [60] and the value for the unmatched sample is 0.72. In summary, the results in Table 5 provide evidence of the reliability of the model that we selected and that matching significantly improved covariate balance.

**Table 5. Covariate balance for the good vision and poor vision groups before and after MDM.**

| Variables | Unmatched sample | | | | Matched sample | | | |
|---|---|---|---|---|---|---|---|---|
| | T | C | T-value | Stand. Diff. % | T | C | T-value | Stand. Diff. % |
| Age | 34.06 | 50.00 | -9.58 | -133.80 | 34.06 | 36.26 | -0.39 | -18.00 |
| Area of cultivation | 3.20 | 4.37 | -3.08 | -40.50 | 3.20 | 3.41 | -0.78 | -7.20 |
| Education | 3.10 | 2.34 | 1.72 | 24.50 | 3.10 | 2.70 | 1.26 | 13.70 |
| Household size | 5.29 | 5.34 | -0.18 | -2.50 | 5.29 | 5.19 | 0.49 | 4.60 |
| Test statistics | Unmatched | | | | Matched | | | |
| Propensity score $R^2$ | 0.24 | | | | 0.01 | | | |
| LR chi$^2$ | 74.73 | | | | 6.06 | | | |
| P>chi$^2$ | <0.01 | | | | 0.19 | | | |
| Mean Bias | 48.60 | | | | 10.90 | | | |
| Rubin's B | 130.10 | | | | 26.00 | | | |
| Rubin's R | 0.72 | | | | 1.64 | | | |

Mean values for the good vision group (T) and the poor vision group (C). Standardized differences are in percent. Rubin's B is the absolute standardized difference of the means of the propensity score in the good vision and poor vision groups (unmatched and matched). Rubin's R is the ratio of the good vision to poor vision variances of the propensity scores. Rubin's B is good if < 25 and Rubin's R is good if >0.5 and <2.0.N matched sample = 255. N unmatched sample = 260. Source: calculated by the authors.

**Table 6. ATT comparison between good vision and poor vision groups with PSM and MDM.**

|  | Treatment | Control | ATT | SE | T-value |
|---|---|---|---|---|---|
| Mahalanobis Distance Matching | 179 | 76 | 632.58 | 287.98 | 2.20 |
| Kernel Density Matching | 179 | 76 | 627.20 | 329.10 | 1.91 |
| Nearest Neighbor Matching | 179 | 40 | 589.38 | 445.96 | 1.32 |
| Sensitivity analysis |  |  |  |  |  |
| E-value: 1.94 |  |  |  |  |  |
| E-value CI: 1.24 |  |  |  |  |  |
| Critical level hidden bias: 1.25 |  |  |  |  |  |

Total observations are 260, 5 drop out when enforcing area of common support. Source: Calculated by the authors.

Table 6 displays the main results from MDM, KBM and NNM. The ATT shows that farmers in the good vision group have net contribution margins that are 632.58 USD higher on average when compared to the poor vision group. The results from KBM are very similar with 627.20 USD/year, though with a lower statistical significance level. The NNM method gives us an ATT of 589.38 USD/year, which is slightly lower than for the other two methods. As opposed to KBM and MDM, NNM does not match on all controls which reduces sample size and inflates the Standard Error. In summary, the results obtained by all three methods are quite close to each other, and taken together give evidence of a positive ATT in the range of 589–632 USD/year associated with having at least intact visual functions in regards to everyday tasks.

## 5.2. Robustness checks

To assess the robustness of our results we apply two slightly different thresholds of belonging to the good vision group. For the lower bound comparison group, we lower the threshold of belonging to the good vision group to all average VA scores that are bigger or equal to 0.45. For the upper bound comparison group we raise this threshold to an average VA score of both eyes bigger or equal to 0.75. The results are displayed in Table 7. For the lower bound comparison group, the results from MDM are robust to the main estimations with an ATT of 679.00 USD/year. For the NNM in the lower bound comparison group, the control group is reduced to only 28 observations which inflate the standard errors and the ATT is not statistically significant. The upper bound comparison group is more balanced in terms of observations in the

**Table 7. ATT's for MDM, KBM and NNM with upper and lower bound comparison groups.**

|  | Treated | Controls | ATT | SE | T-value |
|---|---|---|---|---|---|
| Lower bound comparison group (visus≥0.45) |  |  |  |  |  |
| Mahalanobis Distance Matching | 222 | 38 | 679.00 | 287.26 | 2.36 |
| Kernel Density Matching | 222 | 38 | 273.58 | 518.70 | 0.53 |
| Nearest Neighbor Matching | 222 | 28 | 135.34 | 707.29 | 0.19 |
| Upper bound comparison group (visus≥0.75) |  |  |  |  |  |
| Mahalanobis Distance Matching | 148 | 108 | 617.16 | 365.44 | 1.69 |
| Kernel Density Matching | 148 | 108 | 682.90 | 337.04 | 2.03 |
| Nearest Neighbor Matching | 148 | 53 | 750.36 | 395.55 | 1.90 |

Total observations are 260, 5 drop out when enforcing area of common support. Source: Calculated by the authors.

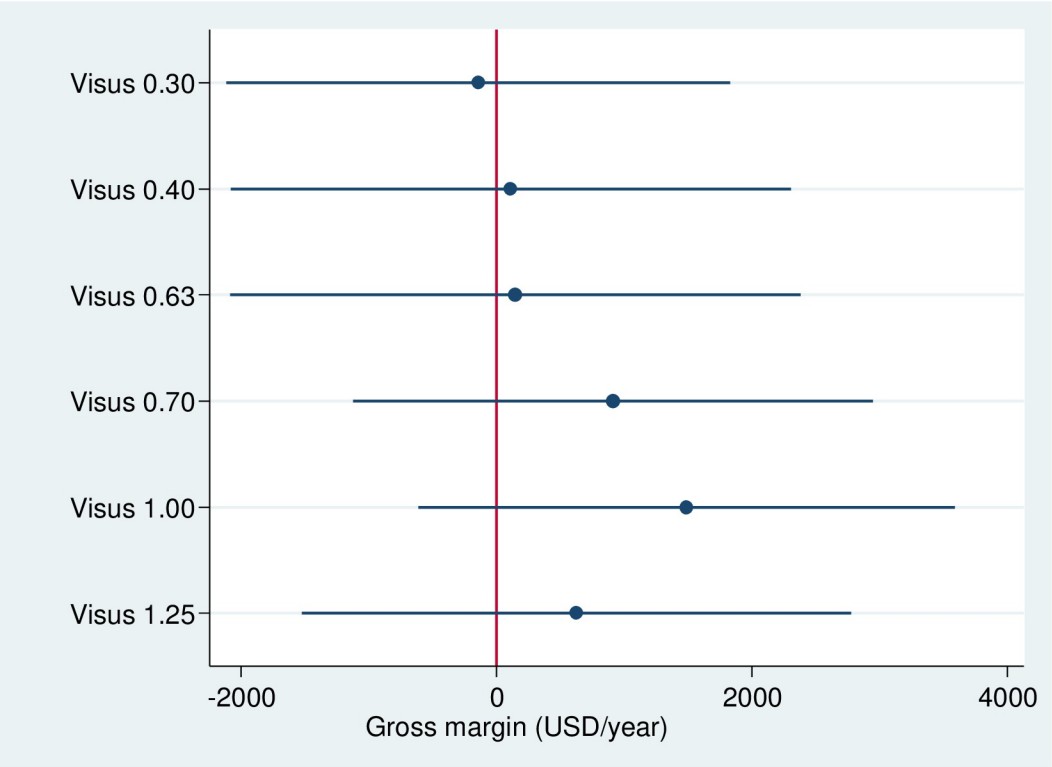

**Fig 4. Results from OLS regression on gross margins, with visual acuity groups as independent factorial variable.** Total observations = 260. Reference group is visual acuity group with a visus of 0. Control variables are not displayed. Source: Own depiction.

control and treatment groups and the calculations yield robust results when compared to the main results.

To look deeper into the effects of the treatment variable, we code visual acuity as a categorical variable, where each line in the vision chart corresponds to one of the following seven visual acuity groups: VA = 0.00; VA = 0.30, VA = 0.40, VA = 0.63, VA = 0.70, VA = 1.00, VA = 1.25. We conduct a regression with this factorial variable, holding all other control variables constant. Fig 4 displays the regression results where the group with a visus of 0 is the reference group. We can observe a stepwise increase in farm profitability up to the visual acuity group with a visus 1.00. The highest increase in farm profitability over the 0 visus group is observed in the group with a visus of 1.00. This advantage over the 0 visus group drops slightly in the group with the highest visus of 1.25. In general, we can observe a near linear increase of farm productivity with visual acuity, which confirms our results from the propensity score matching. Also, the difference between the first three groups is not very strong. But with a visual acuity score of 0.7 we see a sharp increase in contribution margins. Since our results are robust at the upper and lower bound levels, we can conclude that the VA score of 0.70 is a good indicator to assess the impacts of poor vision on farm profitability.

As an additional robustness check we recalculate the outcome variable gross margins. As an alternative indicator we use single factor productivity in which our only input is cropping area per farm, an approach which can be easily interpreted, understood and calculated [61]. More specifically, we multiply produce per plot by average product price in our sample divided by total crop area per farm for the growing season of 2017/2018 for all arable crops of the farm.

**Table 8. Treatment effects of visual acuity on single factor productivity.**

|  | Treatment | T | C | ATT | SE | T-value |
|---|---|---|---|---|---|---|
| Mahalanobis Distance Matching | visus≥0.70 | 179 | 76 | 243.17 | 74.74 | 3.25 |
| Kernel Density Matching | visus≥0.70 | 179 | 76 | 260.98 | 67.45 | 3.87 |
| Nearest Neighbor Matching | visus≥0.70 | 179 | 40 | 237.75 | 109.00 | 2.18 |
| Mahalanobis Distance Matching | visus≥0.45 | 220 | 24 | 320.39 | 69.12 | 4.63 |
| Kernel Density Matching | visus≥0.45 | 220 | 35 | 320.39 | 67.11 | 4.77 |
| Nearest Neighbor Matching | visus≥0.45 | 220 | 24 | 343.83 | 95.07 | 3.62 |
| Mahalanobis Distance Matching | visus≥0.75 | 148 | 107 | 194.28 | 73.23 | 2.65 |
| Kernel Density Matching | visus≥0.75 | 148 | 107 | 190.77 | 71.05 | 2.68 |
| Nearest Neighbor Matching | visus≥0.75 | 148 | 54 | 182.06 | 96.40 | 1.89 |

The outcome variable single factor productivity is expressed in USD per hectare land and year for all plots of the farm. T = Treatment, C = Control. Source: Calculated by the authors.

Table 8 displays the results for the three visual acuity groups and matching algorithms. The ATT for the MDM is 243.17, which means that on average, a farmer in the good vision group earns 243.17 USD more per hectare. The results across groups and matching algorithms remain robust, with the lowest estimation of 182.06 USD/hectare and the highest estimation of 343.83 USD/hectare. Overall, the results obtained by all three methods and visual acuity thresholds taken together show a gain in single factor productivity in the range of 182.06–343.83 USD/hectare/year associated with having at least intact visual functions in regards to everyday tasks.

The sensitivity analysis was carried out and we calculate a critical value for the Rosenbaum bounds of 1.25 (Table 6). For example, for the impact of good vision on gross margins, the sensitivity analysis suggests that at a level of 1.25, there is no hidden bias due to an unobserved confounder. In other words, if the odds of an individual belonging to the good vision group are 1.25 times higher because of the unobserved covariate, despite being identical on the matched (observed) covariate, there may be a change in inference. We can compare this number to the main observable confounder in our data. The variable *Age* alone explains roughly 25 of the variability in treatment status (See S1 Table). Thus, the unobserved confounder needs to have a bigger influence on treatment selection than the variable *Age*. For example, if we had an unobservable confounder like a genetic prevalence that influences treatment selection, this unobservable confounder needs to be unrealistically high. Thus, we assume that our estimates are robust to such unobservable confounders and that possible departures from randomization in our data are not big enough to explain away the pattern that poor eyesight leads to lower farm profitability. The E-value reported in Table 6 supports this result. The observed treatment effect could be explained away by an unmeasured confounder that was associated with both the treatment and the outcome by an effect of 2-fold each, above and beyond the measured confounders. The results have to be taken with caution since we cannot exclude the possibility of potential confounders which influence vision status and gross margins. This can lead to an upward-biased estimate, but we believe that the breadth and depth of our analyses show a clear association between economic farm performance and visual acuity.

## 6. Conclusion

Our study presents first results on the impacts and pathways of visual acuity on economic farm performance in rural Cambodia. We aim to present estimations on the maximum

achievable treatment effect, i.e. to estimate how much profit is forgone because farmers are disadvantaged in managing their farms because of reduced visual functions.

Almost 30 percent of farmers in our sample suffer from poor vision. Furthermore, if a farm manager moves from poor vision to good vision, her gross margins would increase on average by around 630 USD per year. This effect is particularly outstanding considering the Cambodian gross national income per capita (GNI) is 1,380 USD [62]. The result is simple, as is the cure for the problem: glasses. With the help of glasses, a farm manager can potentially switch from poor vision to good vision instantly. According to our data, the economic benefits from this simple intervention can be enormous.

It is questionable if a real-world intervention would deliver a treatment effect of this size, because behavioral aspects would most likely reduce the treatment effect. For example, if a participant would be prescribed glasses, she perhaps wouldn't wear glasses for all activities or couldn't use them for all activities equally. Wearing glasses in field work under direct sun could be practiced less if irritations like fogging and blurred vision due to sweat and dust outweigh the advantages of wearing glasses. Thus, we present estimations on the maximum achievable effect, against which real-world interventions can be measured.

It is clear though, that access to modern optometric services generate high returns on human capital with long lasting effects on educational attainment for example. We provide a framework that shows a variety of effects of poor vision on agricultural profitability. Despite the magnitude of the problem and its relatively cheap solution, which is modern eye care, the relationship between myopia and economic farm performance has received extremely little attention from development actors and researchers alike. Farm managers in the global south are continuously challenged with many technology adoption issues. To make sound management decisions they require an intact visual system. Our results show that there are important linkages between agriculture and public health and that there is a need for more collaboration across the agricultural and public health sectors to address the negative impacts of ill-health on agricultural profitability.

In future research, a better identification of the causal relationships between myopia and farm profitability can be established by collecting longitudinal data. Future research should investigate (1) which entrepreneurial activities are most affected by poor vision and (2) which steps need to be taken to drive the usage of glasses. A repeated measure within-subjects design, e.g. a controlled experiment that applies pre- and post-measurement in relation to the treatment of glasses or contact lenses would be optimal for determining the causal effect of visual acuity on the economic performance of farms of smallholders.

## Supporting information

**S1 File. Symptoms and limitations imposed on rural dwellers by poor visual acuity.**
Adapted from Kandel et al. (2018).
(DOCX)

**S2 File. Inclusivity in global research.**
(DOCX)

**S1 Table. Logit model of eyesight with age as only predictor.**
(DOCX)

## Author Contributions

**Conceptualization:** Frederik Sagemüller, Selina Bruns.

**Data curation:** Selina Bruns.

**Formal analysis:** Frederik Sagemüller.

**Funding acquisition:** Oliver Mußhoff.

**Methodology:** Frederik Sagemüller.

**Software:** Frederik Sagemüller.

**Supervision:** Oliver Mußhoff.

**Writing – original draft:** Frederik Sagemüller.

**Writing – review & editing:** Selina Bruns.

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
