## [Decision Letter · Decision Letter 0]

16 May 2022

PONE-D-22-08676The effect of poor vision on economic farm performance: Evidence from rural CambodiaPLOS ONE

Dear Dr. Sagemüller,

Thank you for submitting your manuscript to PLOS ONE. After careful consideration, we feel that it has merit but does not fully meet PLOS ONE’s publication criteria as it currently stands. Therefore, we invite you to submit a revised version of the manuscript that addresses the points raised during the review process.

We look forward to receiving your revised manuscript.

Kind regards,

László Vasa, PhD

Academic Editor

PLOS ONE

Journal Requirements:

2. For studies reporting research involving human participants, PLOS ONE requires authors to confirm that this specific study was reviewed and approved by an institutional review board (ethics committee) before the study began. Please provide the specific name of the ethics committee/IRB that approved your study, or explain why you did not seek approval in this case.

4. Please include a complete copy of PLOS’ questionnaire on inclusivity in global research in your revised manuscript. Our policy for research in this area aims to improve transparency in the reporting of research performed outside of researchers’ own country or community. The policy applies to researchers who have travelled to a different country to conduct research, research with Indigenous populations or their lands, and research on cultural artefacts. The questionnaire can also be requested at the journal’s discretion for any other submissions, even if these conditions are not met.  Please find more information on the policy and a link to download a blank copy of the questionnaire here: https://journals.plos.org/plosone/s/best-practices-in-research-reporting. Please upload a completed version of your questionnaire as Supporting Information when you resubmit your manuscript.

“This work was supported by the Bundesministerium für Wirtschaftliche Zusammenarbeit und Entwicklung, BMZ, Federal Ministry of Development Cooperation [grant number 81180342].”

7. Please include a caption for figure 1, 2 and 3.

Reviewers' comments:

Reviewer's Responses to Questions

**Comments to the Author**

1. Is the manuscript technically sound, and do the data support the conclusions?

Reviewer #1: Partly

Reviewer #2: Partly

2. Has the statistical analysis been performed appropriately and rigorously? 

Reviewer #1: Yes

Reviewer #2: Yes

3. Have the authors made all data underlying the findings in their manuscript fully available?

Reviewer #1: Yes

Reviewer #2: Yes

4. Is the manuscript presented in an intelligible fashion and written in standard English?

Reviewer #1: Yes

Reviewer #2: Yes

5. Review Comments to the Author

Reviewer #1: The topic is very interesting and also relevant. The elaboration and the applied methodology look accurate. There is still a formal problem: there are no source indications neither under the tables in the paper, nor under the figures at the end of the text. The indication of data sources is an essential requirement even in case of field research data as well. The numbering of figures is not evident, there are three figures indicated as “Figure 1”, then the last one as “Figure 4”. The bibliographic review could also be broadened, with the view of focusing on the international context, like examples of other Asian countries especially in ASEAN region, as Vietnam or Laos. After the mentioned improvements were done, the paper can be recommended to be accepted and published.

Reviewer #2: The paper focuses on a topic with rather unique approach. The correlations of vision problems and farm management is a less researched field so no doubt, the paper is original. Regarding the context, it is obvious that Cambodia is just a kind of case study which allows us to estimate the seriousness of the problem in other developing countries with low level of healthcare.

The statistical analysis uses the appropriate and well.selected toolset and completed well, however, despite most elements of the sampling, dataset and statistical tools are described, the general introduction of the methodology is missing. Which methods were used, based on what assumptions and who else used it for similar purposes? These questions should be answered at the beginning of the methodological parts.

The introductory chapter highlights the reason and context of the research well, and also includes some literature review. However, this latter is far not enough. I recommend to write a separate literature review chapter where the related sources are processed analytically, critically and in a comparative way.

6. PLOS authors have the option to publish the peer review history of their article (what does this mean?). If published, this will include your full peer review and any attached files.

Reviewer #1: No

Reviewer #2: No

---

## [Author Response · Author response to Decision Letter 0]

22 Jul 2022

Point-to-point response to reviewer #1 

Regarding the original research article

“The effect of poor vision on economic farm performance: Evidence from rural Cambodia.”

Dear reviewer #1,

Thank you very much for your helpful feedback. We are very grateful for the time and effort that you have devoted to our paper, and we are confident that our revisions and responses below address your concerns. However, we stand ready to make any additional changes that you deem necessary. Please find below our detailed responses.

The topic is very interesting and also relevant. The elaboration and the applied methodology look accurate. 

Thank you very much for this positive comment!

There is still a formal problem: there are no source indications neither under the tables in the paper, nor under the figures at the end of the text. The indication of data sources is an essential requirement even in case of field research data as well. 

Thank you very much for this information; we added the source to all tables and figures in the manuscript.

The numbering of figures is not evident, there are three figures indicated as “Figure 1”, then the last one as “Figure 4”. 

Thank you very much for detecting this error. We adjusted all captions.

The bibliographic review could also be broadened, with the view of focusing on the international context, like examples of other Asian countries especially in ASEAN region, as Vietnam or Laos. 

Thank you very much for this comment. This topic came up with the other reviewer as well. We decided to add chapter “2. Literature review”, where we include a discussion on 15 more research articles, including studies on the prevalence of vision impairments from Cambodia’s neighboring countries Lao PDR, Myanmar, Thailand and Vietnam. For reasons of brevity we don’t include the chapter here, but kindly refer to the literature review beginning on page 4 of the manuscript.

After the mentioned improvements were done, the paper can be recommended to be accepted and published.

Thank you very much for the positive review, it is very much appreciated.

Point-to-point response to reviewer #2 

Regarding the original research article

“The effect of poor vision on economic farm performance: Evidence from rural Cambodia.”

Dear reviewer #2,

Thank you very much for your helpful feedback. We are very grateful for the time and effort that you have devoted to our paper, and we are confident that our revisions and responses below address your concerns. However, we stand ready to make any additional changes that you deem necessary. Please find below our detailed response.

Reviewer #2: 

The paper focuses on a topic with rather unique approach. The correlation of vision problems and farm management is a less researched field so no doubt, the paper is original. Regarding the context, it is obvious that Cambodia is just a kind of case study which allows us to estimate the seriousness of the problem in other developing countries with low level of healthcare.

Thank you very much for this comment, we are very glad to receive this positive comment. 

The statistical analysis uses the appropriate and well selected toolset and completed well, however, despite most elements of the sampling, dataset and statistical tools are described, the general introduction of the methodology is missing. Which methods were used, based on what assumptions and who else used it for similar purposes? These questions should be answered at the beginning of the methodological parts.

Thank you very much for this comment. We added a paragraph to the methods section, were we introduce the methodology and in which contexts it is used. We also refer to several studies that evaluate impacts on agricultural production which use these methods. On page 13, we added following sentences to the first paragraph: 

An important problem of causal inference is how to estimate treatment effects in observational studies, where (like an experiment) a group of units is exposed to a well-defined treatment, but (unlike an experiment) no systematic methods of experimental design are used to maintain a control group(46). To circumvent this problem, we apply Mahalanobis distance matching (MDM) and propensity score matching (PSM). Matching involves pairing treatment and comparison units that are similar in terms of their observable characteristics. These matching methods have become popular in impact evaluations and are used in a variety of fields, including to assess impacts related to agricultural production (47–51).

The introductory chapter highlights the reason and context of the research well, and also includes some literature review. However, this latter is far not enough. I recommend to write a separate literature review chapter where the related sources are processed analytically, critically and in a comparative way.

Thank you very much for this comment. As you suggested, we included “Section 2. Literature review” to the manuscript. In this literature review, we discuss 15 studies on the global prevalence of vision impairments and their impacts on global productivity. We critically review the research gaps, with a special attention on the impacts of visual impairments on agricultural productivity. For reasons of brevity we don’t include the chapter here, but kindly refer to the literature review beginning on page 4 of the manuscript.

---

## [Decision Letter · Decision Letter 1]

22 Aug 2022

The effect of poor vision on economic farm performance: Evidence from rural Cambodia

PONE-D-22-08676R1

Dear Dr. Sagemüller,

We’re pleased to inform you that your manuscript has been judged scientifically suitable for publication and will be formally accepted for publication once it meets all outstanding technical requirements.

Kind regards,

László Vasa, PhD

Academic Editor

PLOS ONE

Additional Editor Comments (optional):

Reviewers' comments:

Reviewer's Responses to Questions

**Comments to the Author**

1. If the authors have adequately addressed your comments raised in a previous round of review and you feel that this manuscript is now acceptable for publication, you may indicate that here to bypass the “Comments to the Author” section, enter your conflict of interest statement in the “Confidential to Editor” section, and submit your "Accept" recommendation.

Reviewer #1: All comments have been addressed

Reviewer #2: All comments have been addressed

2. Is the manuscript technically sound, and do the data support the conclusions?

Reviewer #1: Yes

Reviewer #2: Yes

3. Has the statistical analysis been performed appropriately and rigorously? 

Reviewer #1: Yes

Reviewer #2: Yes

4. Have the authors made all data underlying the findings in their manuscript fully available?

Reviewer #1: Yes

Reviewer #2: Yes

5. Is the manuscript presented in an intelligible fashion and written in standard English?

Reviewer #1: Yes

Reviewer #2: Yes

6. Review Comments to the Author

Reviewer #1: All the previous suggestions and critical observations have been fixed, so the paper is recommended to be accepted.

Reviewer #2: The authors accepted my critics and improved their paper based on my comments and recommendations. The paper represents a higher scientific standard now, I find it eligible for publication without any further changes.

7. PLOS authors have the option to publish the peer review history of their article (what does this mean?). If published, this will include your full peer review and any attached files.

Reviewer #1: No

Reviewer #2: No

---

## [Editor Report · Acceptance letter]

30 Aug 2022

PONE-D-22-08676R1 

The effect of poor vision on economic farm performance: Evidence from rural Cambodia 

Dear Dr. Sagemüller:

I'm pleased to inform you that your manuscript has been deemed suitable for publication in PLOS ONE. Congratulations! Your manuscript is now with our production department. 

Kind regards, 

on behalf of

Prof. Dr. László Vasa 

Academic Editor

PLOS ONE